# The Healthcare Experience of Autistic Patients in Orthopaedic Surgery and Closely Related Fields: A Scoping Review

**DOI:** 10.3390/children10050906

**Published:** 2023-05-22

**Authors:** Steven D. Criss, Shravya Kakulamarri, Raylin F. Xu, Maya Fajardo, Tamra Keeney, Dorothy W. Tolchin, Collin J. May

**Affiliations:** 1Harvard Medical School, Boston, MA 02115, USA; steven.criss@childrens.harvard.edu (S.D.C.);; 2University of Texas Southwestern Medical Center, Dallas, TX 75390, USA; 3Department of Orthopedic Surgery, Boston Children’s Hospital, Boston, MA 02115, USA; 4Mongan Institute, Massachusetts General Hospital, Boston, MA 02114, USA; 5Division of Palliative Care and Geriatric Medicine, Massachusetts General Hospital, Boston, MA 02114, USA; 6Department of Physical Medicine and Rehabilitation, Mass General Brigham, Boston, MA 02114, USA; 7Spaulding Rehabilitation Hospital, Charlestown, MA 02129, USA

**Keywords:** autism spectrum disorder, autism spectrum conditions, orthopaedic surgery, physical medicine and rehabilitation, physical therapy, occupational therapy, patient experience, quality improvement

## Abstract

Orthopaedic and related care has the potential to present unique obstacles for patients with a range of autism manifestations. In this review, we aim to describe and analyze the literature on autistic patients’ experience within orthopaedics and closely related fields. This literature search utilized the PubMed, Embase, and Cumulative Index to Nursing and Allied Health Literature databases. Three major concepts were built into the search terms: (1) patients on the autism spectrum; (2) patient experience; and (3) movement sciences, including orthopaedics, physical medicine and rehabilitation (PM&R), occupational therapy (OT), and physical therapy (PT). Our search yielded 35 topical publications, with the major topic areas addressed as follows: (1) clinical and perioperative management, (2) therapy interventions, (3) participation in exercise and social play, (4) sensory management and accommodations, (5) caregiver/parent training and involvement in care, (6) healthcare needs and barriers to care, and (7) utilization of technology. In the current literature, there are no studies that attempt to directly assess autistic patient experience with care practices and clinical environments in orthopaedics. Rigorous, direct examination of the experience of autistic patients within clinical orthopaedic settings is urgently needed to address this gap.

## 1. Introduction

Autistic patients face considerable challenges when navigating the healthcare system. There are several cardinal features of autism spectrum disorder (ASD)—namely, difficulty with social communication and reciprocal social interaction, restricted or repetitive behaviors and/or interests, and atypical sensitivity to stimuli (Table 1) [1,2]. Management of such features is often complicated in the medical setting by the rigidity of a largely volume-based economic model and numerous yet unavoidable discomforts of a clinical encounter, which can lead to profound stress and anxiety for autistic patients [2,3,4,5,6,7,8]. Autistic patients also commonly experience comorbidities that augment their exposure to the healthcare system, including intellectual/developmental disorders, motor abnormalities, medical disorders, psychiatric disorders, and behavioral issues (Table 1) [1,2,9,10,11,12,13,14,15,16,17,18,19].

With the prevalence of ASD in children in the U.S. estimated to be between 1.7% to 2.5%, providers in every pediatric specialty should consider how aspects of their practice may differentially impact autistic patients [3,20,21,22,23,24]. This is particularly important within specialties where using individualized communication and interaction strategies for a patient is challenging due to the need for rapid decision making, uncomfortable medical devices, or other factors intrinsic to the specialty. Orthopaedic surgery is one such field that may present substantial obstacles to autistic patients who experience social, behavioral, or sensory issues. Many aspects of orthopaedic care—physical manipulation during orthopaedic exam, inflexible requirements of bracing and therapy protocols, and physically uncomfortable or painful elements of surgery and casting—may elevate distress experienced by autistic patients beyond that of non-autistic peers. Given that autistic patients may also be managing behavioral, psychiatric, and intellectual challenges, ensuring attention is paid to the amount of elevated distress in these individuals is all the more crucial [11,13,18,25,26]. Furthermore, the stress of the various manifestations of autism and its associated comorbidities reaches beyond the patients themselves and plays a role in the well-being of their caregivers [27].

Understanding the experience of autistic patients who receive orthopaedic care and identifying existing interventions or accommodations is an important first step toward ensuring accessible, comfortable care. A recent review of strategies for improving autistic patient experience in the emergency department revealed numerous potential interventions specific to the emergency care setting [28]. In this current review, we conducted a systematic literature search of orthopaedics and closely related fields—physical medicine and rehabilitation (PM&R), occupational therapy (OT), and physical therapy (PT)—to inform orthopaedic care more broadly for a patient population generally known to experience ongoing challenges throughout the healthcare system [7,8].

## 2. Materials and Methods

The existing evidence regarding ASD patient experience in orthopaedics and its partnering fields has yet to be assessed in a systematic and exhaustive manner, leading our team to pursue this review [29,30]. Our search of the literature utilized the PubMed, Embase, and Cumulative Index to Nursing and Allied Health Literature (CINAHL) databases. Three major concepts were built into the search terms: (1) patients on the autism spectrum; (2) patient experience; and (3) movement sciences, including orthopaedics, PM&R, OT, and PT. The query formulas used are provided in Appendix A. We limited the queries to English-language publications but did not limit by date of publication. We used the Preferred Reporting Items for Systematic Reviews and Meta-Analyses (PRISMA) guidelines to perform our literature review [31]. The literature search used includes articles published as of 3 March 2023.

Publications were imported into Covidence, a systematic review management site (www.covidence.org; accessed on 2 May 2022), and screened independently by three members of the team to decide which were relevant for inclusion (S.D.C., S.K., R.F.X.). Inclusion criteria were as follows: autistic patients as part of the patient sample, patient experience-related outcomes (e.g., satisfaction, qualitative response to intervention, among others), and relation to the movement sciences. There were no specific exclusion criteria. We did not limit inclusion by country of publication, country of patient sample, age of patient sample, or research methodology. Inclusion and exclusion decisions were reviewed and approved by the senior author (C.J.M.).

Publication types included original research articles, qualitative reports, case studies/series, and reviews. Our initial search yielded 114 unique publications for consideration. The screening procedure resulted in 35 relevant articles for inclusion in this review, which were then categorized by their subject matter into seven major topics by the screening team and senior author (S.D.C., S.K., R.F.X., C.J.M.). The collection of data was performed independently by the first author (S.D.C.) and reviewed by the other two screening authors (S.K., R.F.X.) as well as the senior author (C.J.M.). Outcomes included patient experience-related data and interventions or factors that could affect experience.

## 3. Results

The 35 selected publications from the literature search (Figure 1) were published between 2004 and 2023. The most common research methodologies used were surveying (n = 12), mixed-methods or other analysis of a non-randomized intervention (n = 5), literature review (n = 5), qualitative/thematic analysis of participant interviews (n = 4), and randomized controlled trial (n = 4). Only 3 of the publications came from the orthopaedics literature, while 16 were from the OT, PT, and/or PM&R literature (n = 11, 4, and 2, respectively; PT and PM&R share one publication), with the remaining 16 split among pediatrics, neurology, psychiatry/psychology, and other fields. Other characteristics of the included publications are presented in Table 2. Within the patient samples of the included studies were numerous conditions that are commonly associated and overlap with ASD: neurodevelopmental disorders, motor impairment, intellectual disability, medically complicated/hospitalized, genetic syndromes, psychiatric conditions, and behavioral issues. The age ranges of patients in the included studies were broad, including preschool-age children to adolescents and adults.

### 3.1. Relevant Topic Areas

#### 3.1.1. Clinical and Perioperative Management

The three publications from the orthopaedic literature addressed clinical and perioperative management for autistic patients. Two review articles provided recommendations to create a positive healthcare interaction for autistic patients in orthopaedic surgery, while acknowledging the lack of research into the unique needs of autistic patients in orthopaedics [38,48]. These recommendations, however, do not draw on evidence specific to orthopaedics and, consequently, provide general best practices applicable to most procedure-based settings. In the third article, Valencia et al. investigated regional blocks with bupivacaine only (control) versus bupivacaine plus dexamethasone (experimental) in 39 patients with chronic musculoskeletal conditions undergoing orthopaedic procedures (ASD patient n = 7, not separately discussed) [62]. Notable experience-related results included improvements in pain medication use, patient and parent satisfaction, and anxiety level over future procedures [62]. The authors note that the improved pain management allowed the majority of the procedures to be performed in the outpatient setting, reducing family disruption associated with inpatient hospitalizations [62].

#### 3.1.2. Therapy Interventions

A variety of therapeutic interventions provided to autistic patients are described in the literature (e.g., animal-assisted therapy, aquatic environment therapy, music therapy, and movement-based therapy), most of which report adequate feasibility and acceptability for patients, caregivers or parents, and/or therapists.

Equine-assisted therapy is discussed by two studies (a randomized study of 23 patients and a qualitative analysis of 4 patients) as having promising results in terms of satisfaction and adherence to the intervention, as well as improvements in occupational and emotional performance qualitatively credited to the working relationship developed between horse and patient [47,54]. A study of canine-assisted therapy for 44 autistic children showed longitudinal improvement in communication skills, social relations, and engagement levels [33].

Therapy in an aquatic setting was shown to have benefits for both physical competence and social interaction [41]. A review of music therapies for children with ASD acknowledged a lack of sufficient evidence in this population, but described how music-based interventions that provide patients an opportunity to actively engage (i.e., singing, music-making, and synchronized rhythmic actions) demonstrate the most potential for benefit [59]. Similarly, in a systematic review, movement-based interventions were shown to have potential for benefit to body structure and function in autistic patients, but results were mostly not significant [35].

#### 3.1.3. Participation in Exercise and Social Play

Participation in social play and exercise are documented challenges for many autistic patients [43,45]. In a study of participation patterns for 144 preschool-aged children, those with ASD participated in fewer activities, with parents most commonly citing behavioral difficulties and sensory issues as major barriers [45]. In a study of participation in exercise activities with 83 autistic children, additional identified barriers included financial burden and lack of opportunity [50].

Studies have also tested ways to increase the physical activity of ASD patients. In one study aimed at increasing athletic group play, three autistic children were successfully taught athletic skills using clear, direct instructions and positive reinforcement from a physical therapist [51]. Additionally, a review of active video games (exergames) noted that, while evidence is limited, exergames could be a more enjoyable and comfortable alternative activity for autistic children [46]. Finally, a pilot study of five autistic children used a play-based intervention pairing patients with typically developing playmates, showing positive feasibility and improvements in social play [42]. These strategies have improved activity participation rates and comfort while learning new skills, and may provide guidance for how to boost adherence and visit participation in orthopaedics.

#### 3.1.4. Sensory Management and Accommodations

Sensory management has been shown to affect the acceptability of therapy and various domains of development. In a survey of 211 occupational therapists, 98% recommended using sensory-based approaches for autistic patients (e.g., sensory diets, sensory stimulation with weight or pressure, and sensory integration in a specialized environment), though they would only recommend this for 57% of the actual autistic patients they treat [61]. Further support for sensory integration came from a study of 24 autistic children that showed integration strategies improved cognitive, emotional, social, and motor development [65,66].

Moreover, in a real-world occupational study of 50 working adults with ASD, positive associations between job satisfaction and physical comfort, as well as supportive social environments, were found [55]. An occupational training model designed to teach self-regulated problem-solving skills proved helpful for 25 autistic individuals as they transitioned to employment through improving their occupational performance and goal attainment [37]. Assessing the sensory burden of various orthopaedic care settings could help guide implementation of accommodations for autistic patients in the form of sensory-friendly changes to the physical environment or making sensory-based therapeutic devices and activities available.

#### 3.1.5. Caregiver/Parent Training and Involvement in Care

Several articles in the present review addressed the importance of involving parents and caregivers in therapy and soliciting their perspectives in the care of autistic patients [27]. Involvement of caregivers was key to the success of an occupational performance intervention featuring parent coaching and social support studied in a randomized trial of 36 parents of autistic children [52]. Parents of children with ASD (n = 35) were enrolled in another randomized intervention to build their resilience, resulting in gains in stress management and parent-reported child functioning [58]. Additionally, therapy aimed at parent–child interaction was also found to be helpful for reducing disruptive behavior in a case study of a single patient with ASD [32].

Several other studies reported on caregiver perspectives of the treatment strategies used for their autistic children. Qualitative thematic analysis of 13 parents of autistic children showed the most important parental needs for enabling their child’s participation included tailored interventions and support for parents in finding appropriate leisure activities [36]. These parents reported supporting their child’s participation through promoting resiliency, attaining proper fit between their child and the activity’s characteristics, and providing inclusive opportunities [36]. A qualitative analysis of 25 parents found the determining factors in their child’s participation were the sensory characteristics of the physical environment and the inclusiveness of the activity’s social environment [34]. Regarding parent satisfaction with the treatment process, 23 parents of children with developmental disabilities (more than half of which had a confirmed ASD diagnosis) were broadly satisfied with primary care and rehabilitation teams, yet were less satisfied with specialist and social care providers [53]. Finally, a large study of caregivers for 1296 children with Fragile X syndrome found that patients with more severe ASD features had higher utilization of OT, speech-language therapy, and behavioral therapy [49].

#### 3.1.6. Healthcare Needs and Barriers to Care

A patient’s healthcare needs and the level of access to services that address those needs can affect their experience of care. A survey of 40 autistic adults without intellectual disability demonstrated that the most common barriers to receiving needed health services were not knowing where to find help (66%), overwhelming steps to receive care (53%), and negative experiences with health professionals (47%) [63]. Cost of services has also been a reported barrier to care, as discontinuation of rehabilitation was most attributed to high cost in a study of 225 children with developmental disorders (164 diagnosed with ASD) [56]. An analysis of service data by insurance type from 113 caregivers of children with ASD revealed that access to care had a significant positive correlation with satisfaction with the payer (public or private) and a negative correlation with parent stress [64].

Providers were surveyed in a study of 71 OT practitioners to determine the challenges of managing puberty in autistic adolescents [44]. The most commonly noted challenges were emotional regulation, behavioral management, social participation, and coping with puberty; the interventions used most commonly to address these challenges were behavioral strategies, social learning, parent training, and sensory processing integration [44].

Perhaps one of the most direct investigations of the healthcare experience of autistic patients came from an effort to establish a clinical pathway for hospitalized autistic patients that would ameliorate the more difficult aspects of their admission, though details were only published in abstract form [40]. The pathway required the patients’ care teams to directly address ASD-related needs on a daily basis, which resulted in unanimous support among patient caregivers [40].

#### 3.1.7. Utilization of Technology

Interventions utilizing technology to provide therapy or advance care plans at home are not yet widely available. A qualitative study used focus group interviews of occupational therapists to inform plans for an application that aims to help children with ASD and their families regulate sensory processing issues [57]. A report on the use of telehealth OT and PT interventions for autistic patients during the COVID-19 pandemic demonstrated positive preliminary acceptability and satisfaction [60]. Contrarily, a separate study of predictors of satisfaction with ASD services during the COVID-19 pandemic revealed that occupational, behavioral, and speech/language therapy delivered via telehealth earned lower satisfaction ratings compared to in-person sessions [39]. Understanding how autistic patients best utilize and experience these new technologies and virtual services will be critical to their successful implementation in this population.

## 4. Discussion

In this review of the orthopaedics, PM&R, OT, and PT literature, we describe 35 publications that address the experience of autistic patients being cared for within these fields. The selected publications primarily report on ASD patient experience indirectly through assessing satisfaction with implemented interventions or describing whether a practice was effective in promoting social and developmental progress. It is notable that none of the publications derived in this comprehensive scoping review directly elicited patient perspectives on what is helpful or challenging about the care experience itself. A summary of key takeaways from the literature, discussed in greater depth below, is included in Figure 2.

Numerous innovative therapy methods for autistic patients have been trialed, though none of these interventions were studied in ASD patients being treated specifically for orthopaedic conditions. The breadth of feasible therapy strategies in different environments should be an encouraging sign that changes to standard practice could be well-tolerated. Identifying the barriers to participation in activities, such as behavioral issues or sensory sensitivities, could help optimize adherence to different aspects of care, such as postoperative therapy regimens or orthopaedic bracing requirements. Elements of interventions to improve participation—group play, active video games, or mobile applications—should be studied in the orthopaedic setting to assess whether they can help improve autistic patient comfort and stress levels.

The limited available literature on optimizing orthopaedic clinical encounters and preventing behavioral issues for patients with ASD highlights a need for advancing discourse in this area. The aspects of orthopedic care most stressful for patients with ASD and their families, the range of ways autistic patients might manifest stress in response to orthopedic interventions, and how orthopaedic providers and clinic spaces should prepare for visits all represent critical gaps in our practical knowledge of caring for autistic patients. One takeaway from the present scoping review is that sensory management strategies are beneficial in OT and PT settings; perhaps these approaches can be tailored for orthopedic settings where there is close physical interaction and maneuvering. Care should be taken to tailor approaches to each patient based on their particular needs and providers should be open-minded about changing strategies if outcomes are not in line with expectations.

A second takeaway from the current literature is the importance of involving caregiver training in treatment plans. Their perspectives can be an irreplaceable source of information about the specific needs that should be addressed in the clinic or operating room, the services that may be most helpful, and the accommodations their child responds to best. The field of orthopaedics should also consider how barriers to accessing care, logistical complexities of receiving care, and prior negative experiences may manifest in practice.

Ample opportunity exists within orthopaedics to study the experience of autistic patients. Using the cardinal features of ASD as a guide and the gaps revealed in the present scoping review, we present several starting points for future study.

○Social participation and communication challenges: Orthopaedics naturally works very closely with other specialties, which, in turn, means patients must be set up for numerous visits with several different providers. The impact of these social and communication demands on autistic patients warrants study. Other social and communication-related topics include how pain and other discomforts are relayed, the duration of appointments, who is allowed in the room during the visit, and the amount of preparation provided for upcoming appointments or procedures [28].○Restricted and/or repetitive behaviors, routines, and activities: Treatment for orthopaedic conditions often involves regimented therapy plans as well as activity and weight-bearing restrictions, which may be difficult for autistic patients who struggle adapting to new routines. A more thorough understanding of how these patients respond to such restrictions could be important to improving their postoperative experience. Restricted diets of autistic patients may predispose them to poor bone mineral density, with a plausibly higher risk for fracture [67,68,69]. Broadening a patient’s diet may be an important part of their recovery and future bone health, but how they are able to tolerate such changes during recovery remains to be seen.○Differences in sensory sensitivity and processing: The sensory demands of various orthopaedic clinical environments are significant. Crowded waiting rooms, bright perioperative spaces with countless audible signals and conversations, an assortment of tapes and leads on the skin, and gowns with stiff fabrics are only some of the more obvious factors that could exacerbate challenges with sensory processing [28]. Certain situations and equipment are also prone to aggravating these ASD features, namely casting and cast saw use, maintaining proper limb positioning with braces, and manipulation during the physical exam. Future research should focus on which of these factors may have the greatest impact on autistic patients and what methods can help alleviate the stress of high sensory burden.

This review has several limitations. Only three studies from the orthopaedic literature qualified for inclusion, preventing us from commenting more thoroughly on the experience of autistic patients specifically in orthopaedics. Additionally, given the overall dearth of experience data in the literature, high-quality evidence was rare in the studies we synthesized in our review. Finally, to maximize the number of publications to draw on, we included work from different countries with different health systems and ASD prevalence, which limits generalization of these findings to any one health system.

## 5. Conclusions

The current published literature in the field of orthopaedics inadequately assesses how autistic patients experience their care. Contributions from the PM&R, OT, and PT literature provide valuable support to this topic, but ultimately fall short of directly addressing which methods of care should be more widely promoted for this patient population and which should be modified. Rigorously examining the experiences of autistic patients within orthopaedics is a much-needed step toward empowering orthopaedists to both optimize care for these patients and serve as allies to successfully navigate medical and surgical care.

## Figures and Tables

**Figure 1 children-10-00906-f001:**
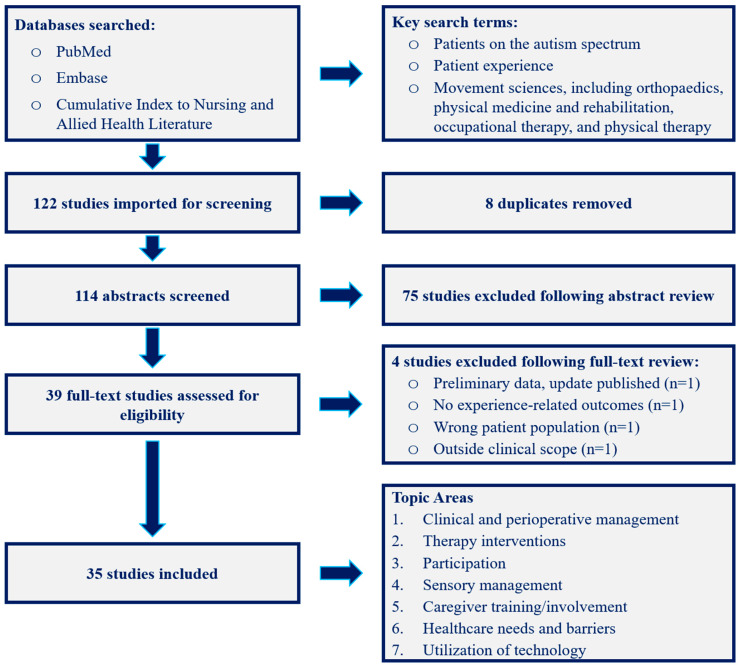
PRISMA diagram, as of 3 March 2023.

**Figure 2 children-10-00906-f002:**
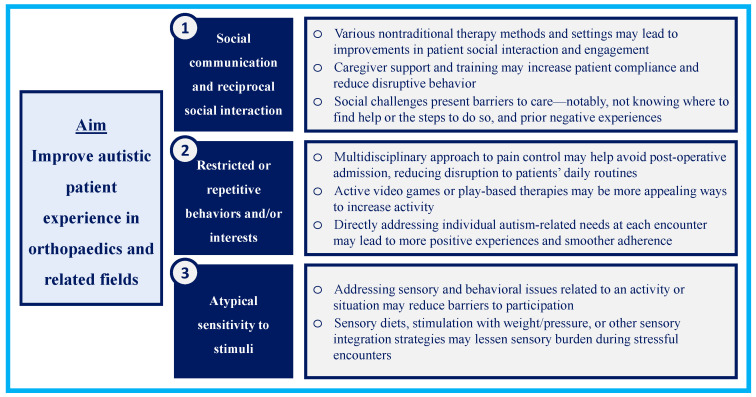
Key takeaways: strategies for improving autistic patient clinical experiences.

**Table 1 children-10-00906-t001:** Features of autism spectrum disorder and related conditions [1,2,9,10,11,12,13,14,15,16,17,18,19].

DSM-5 Diagnostic Criteria	
Persistent deficits in social communication and social interaction across multiple contexts	Deficits in social–emotional reciprocity Deficits in non-verbal communicative behaviors used for social interaction Deficits in developing, maintaining, and understanding relationships
Restricted, repetitive patterns of behavior, interests, or activities	Stereotyped or repetitive motor movements, use of objects, or speech Insistence on sameness, inflexible adherence to routines, or ritualized patterns of verbal or non-verbal behavior Highly restricted, fixated interests that are abnormal in intensity or focus Hyper- or hyporeactivity to sensory input or unusual interest in sensory aspects of the environment
**Commonly associated comorbidities**	
Intellectual/Developmental	Language disorders, language delay Intellectual disability, various causes/types Attention deficit hyperactivity disorder Tourette’s Syndrome
Motor	Delay in motor function; hypotonia; catatonia; difficulty with coordination, movement planning, praxis, gait, and balance
Medical	Gastrointestinal issues (constipation, diarrhea, abdominal pain, gastroesophageal reflux, esophagitis) Deficits in immune function Genetic syndromes Epilepsy
Psychiatric	Depression Anxiety Obsessive compulsive disorder
Behavioral/Personality	Self-injurious behavior Restrictive food intake Aggressive behaviors Various personality disorders

DSM-5 = Diagnostic and Statistical Manual of Mental Disorders, 5th edition.

**Table 2 children-10-00906-t002:** Additional information for included publications.

Author, Year	Title	Design	N	Topic Area
Agazzi et al., 2013 [32]	A Case Study of Parent-Child Interaction Therapy for the Treatment of Autism Spectrum Disorder	Case study/series	1	Caregiver/parent training and involvement in care
Avila-Alvarez et al., 2022 [33]	Changes in social functioning and engagement during canine-assisted intervention for children with neurodevelopmental disorders in the context of an intervention service	Single arm intervention	44	Interventions in therapy provided and therapy environments
Bedell et al., 2011 [34]	Parent Perspectives to Inform Development of Measures of Children’s Participation and Environment	Qualitative/thematic analysis	42	Caregiver/parent training and involvement in care
Cameron et al., 2019 [35]	Movement-based interventions for preschool-age children with, or at risk of, motor impairment: a systematic review	Review	N/A	Participation in exercise and social play
Coussens et al., 2021 [36]	Participation of young children with developmental disabilities: parental needs and strategies, a qualitative thematic analysis	Qualitative/thematic analysis	13	Caregiver/parent training and involvement in care
Dean et al., 2022 [37]	Promoting Career Design in Youth and Young Adults with ASD: A Feasibility Study	Single arm intervention	25	Sensory management and accommodations
Deon Kidd et al., 2018 [38]	Preparing for Autistic Patients in Orthopaedic Surgery	Review	N/A	Clinical and perioperative management
Ferguson et al., 2021 [39]	Predictors of Satisfaction with Autism Treatment Services During COVID-19	Survey	339	Utilization of technology
Fernandes, 2020 [40]	A Clinical Pathway to Improve this Experience of Medically Hospitalized Patients with Autism Spectrum Disorder	Survey	44	Healthcare needs and barriers to care
Gueita-Rodriguez et al., 2021 [41]	Effects of Aquatic Therapy for Children with Autism Spectrum Disorder on Social Competence and Quality of Life: A Mixed Methods Study	Mixed-methods analysis	6	Interventions in therapy provided and therapy environments
Henning et al., 2016 [42]	A pilot play-based intervention to improve the social play interactions of children with autism spectrum disorder and their typically developing playmates	Case study/series	5	Participation in exercise and social play
Kheirollahzadeh et al., 2020 [43]	The Relationship of School Participation with Motor Proficiency and Executive Function in Children with Autism Spectrum Disorder	Cross-sectional (descriptive-analytic) study	52	Participation in exercise and social play
Larson et al., 2021 [44]	Addressing Puberty Challenges for Adolescents with Autism Spectrum Disorder: A Survey of Occupational Therapy Practice Trends	Survey	71	Healthcare needs and barriers to care
LaVesser & Berg, 2010 [45]	Participation Patterns in Preschool Children with an Autism Spectrum Disorder	Survey	144	Participation in exercise and social play
Lima et al., 2020 [46]	Exergames for children and adolescents with autism spectrum disorder: an overview	Review	N/A	Participation in exercise and social play
Mahoney et al., 2021 [47]	Equine-Assisted Activities and Therapies for Adolescents with Autism Spectrum Disorder: The Lived Experience	Qualitative/thematic analysis	4	Interventions in therapy provided and therapy environments
Maloy et al., 2023 [48]	A Practical Guide for Improving Orthopaedic Care in Children with Autism Spectrum Disorder	Review	N/A	Clinical and perioperative management
Martin et al., 2012 [49]	Therapy service use among individuals with fragile X syndrome: findings from a US parent survey	Survey	1296	Caregiver/parent training and involvement in care
Memari et al., 2015 [50]	Children with Autism Spectrum Disorder and Patterns of Participation in Daily Physical and Play Activities	Survey	83	Participation in exercise and social play
Miltenberger & Charlop, 2013 [51]	Increasing the Athletic Group Play of Children with Autism	Case study/series	3	Participation in exercise and social play
Azari et al., 2019 [52]	Contextual Intervention Adapted for Autism Spectrum Disorder: An RCT of a Parenting Program with Parents of Children Diagnosed with Autism Spectrum Disorder	Randomized controlled trial	36	Caregiver/parent training and involvement in care
Pek et al., 2021 [53]	Parent Perceptions of Diagnostic Process and Treatment in Children with Developmental Disabilities	Survey	23	Caregiver/parent training and involvement in care
Peters et al., 2021 [54]	The Feasibility and Acceptability of Occupational Therapy in an Equine Environment for Youth with Autism Spectrum Disorder	Randomized controlled trial	24	Interventions in therapy provided and therapy environments
Pfeiffer et al., 2017 [55]	Impact of person-environment fit on job satisfaction for working adults with autism spectrum disorders	Survey	50	Sensory management and accommodations
Razjouyan et al., 2021 [56]	Dropout from Rehabilitation and Its Associated Factors in Children with Developmental Disabilities in Tehran Rehabilitation Centers	Mixed-methods analysis	225	Healthcare needs and barriers to care
Reis et al., 2021 [57]	Regul-A: A Technological Application for Sensory Regulation of Children with Autism Spectrum Disorder in the Home Context	Qualitative/thematic analysis	4	Utilization of technology
Schwartzman et al., 2021 [58]	Resilience Intervention for Parents of Children with Autism: Findings from a Randomized Controlled Trial of the AMOR Method	Randomized controlled trial	35	Caregiver/parent training and involvement in care
Srinivasan & Bhat, 2013 [59]	A review of “music and movement” therapies for children with autism: embodied interventions for multisystem development	Review	N/A	Interventions in therapy provided and therapy environments
Su et al., 2021 [60]	Short report on research trends during the COVID-19 pandemic and use of telehealth interventions and remote brain research in children with autism spectrum disorder	Survey	N/A	Utilization of technology
Thompson-Hodgetts & Magill-Evans, 2018 [61]	Sensory-Based Approaches in Intervention for Children With Autism Spectrum Disorder: Influences on Occupational Therapists’ Recommendations and Perceived Benefits	Survey	211	Sensory management and accommodations
Valencia et al., 2017 [62]	Enhanced perioperative pain management in children with disabilities undergoing lower extremity orthopedic surgery: Does the addition of steroids prolong the effectiveness of regional blocks?	Randomized controlled trial	39	Clinical and perioperative management
Vogan et al., 2017 [63]	Tracking health care service use and the experiences of adults with autism spectrum disorder without intellectual disability: A longitudinal study of service rates, barriers and satisfaction	Survey	40	Healthcare needs and barriers to care
Young et al., 2009 [64]	Public vs. private insurance: Cost, use, accessibility, and outcomes of services for children with autism spectrum disorders	Survey	107	Healthcare needs and barriers to care
Zawadzka, 2014 [65]	Evaluation of the effectiveness of Sensory Integration and Sherborne Developmental Movement in improving the psychomotor functioning of autistic children	Mixed-methods analysis	24	Sensory management and accommodations

## Data Availability

The data that support the findings of this study are available in the text and Appendix A of this article.

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
