# Peer review of "The Healthcare Experience of Autistic Patients in Orthopaedic Surgery and Closely Related Fields: A Scoping Review"

_children, 2023, doi:10.3390/children10050906_

Round 1

Reviewer 1 Report

The study is interesting and useful as a first step in raising medical assistance awareness of ASD patients. However, I would like to make some considerations.
It should write ASD (Autism Spectrum Disorder) rather than Autism Spectrum Condition, referring to the definition of the DSM-5.
It should pay attention to the insertion of Table 2, which partially overlaps line 141; you could put it directly at the end of the Discussion paragraph rather than in the middle of it.
Furthermore, it would be interesting to verify some characteristics of the ASD patients of the studies considered, for example comorbidity with other neurodevelopmental disorders or genetic conditions or age range, which could to frame more specifically the patients you are considering.
It should increased the introduction with more bibliographic references that describe some important elements in autistic patients. For example, it could considering the various reasons why it is important to adopt certain precautions for these children, not only for their disorder itself, but also for particular elements that can characterize them. There may in fact be specific deficiencies in IQ, behavioral and emotional problems compared to a group of normotypicals, as reported in studies of 2021, and these factors can influence medical interventions. Therefore, ASD and its characteristics can also impact on parenting stress, as demonstrated in a 2016 study, so it may be helpful to consider this family aspect, which in turn can affect the overall experience with clinical and care practices.

Author Response

We greatly appreciate the opportunity to revise our manuscript based on the recommendations of the reviewers. We have addressed each comment point by point below and have noted where changes appear in the clean, non-annotated manuscript by page and line number.

  1. It should write ASD (Autism Spectrum Disorder) rather than Autism Spectrum Condition, referring to the definition of the DSM-5.

We have made this change in our manuscript. We originally wrote autism spectrum conditions because the term “disorder” is considered to have a negative connotation by some. However, we agree with the use of the DSM-5 term in this academic setting.

Page: 1, Line: 34

  1. It should pay attention to the insertion of Table 2, which partially overlaps line 141; you could put it directly at the end of the Discussion paragraph rather than in the middle of it.

We believe this comment refers to Figure 2, as the image overlapped with line 241 in the discussion. We have adjusted the size of the image so that it fits within the width of the text, without overlapping the line number. If we are mistaken in this change, we are happy to make additional edits.

Page: 15, Line: 265

  1. Furthermore, it would be interesting to verify some characteristics of the ASD patients of the studies considered, for example comorbidity with other neurodevelopmental disorders or genetic conditions or age range, which could to frame more specifically the patients you are considering.

We appreciate this recommendation and have added text to our Introduction, as well as added a section on the common comorbidities of autistic patients in Table 1, to better frame our patient population. We have also added more details to our Results summarizing the various characteristics that were common to autistic patients in the studies we included in our literature review, including neurodevelopmental disorders, genetic conditions, and the general age ranges of patients, among others.

Page: 1, Line: 40

Page: 2, Line: 44

Page: 4, Line: 110

  1. It should increased the introduction with more bibliographic references that describe some important elements in autistic patients. For example, it could considering the various reasons why it is important to adopt certain precautions for these children, not only for their disorder itself, but also for particular elements that can characterize them. There may in fact be specific deficiencies in IQ, behavioral and emotional problems compared to a group of normotypicals, as reported in studies of 2021, and these factors can influence medical interventions. Therefore, ASD and its characteristics can also impact on parenting stress, as demonstrated in a 2016 study, so it may be helpful to consider this family aspect, which in turn can affect the overall experience with clinical and care practices.

We have augmented our literature cited in the Introduction in our revised manuscript with added text and a revised version of Table 1 that now addresses common comorbidities associated with autistic patients. Specifically, we have now touched on how autism can be intertwined with intellectual and behavioral issues, as well as the effect this spectrum of conditions can have on the patient’s family. We appreciate the reviewer’s suggestion to include these points as further justification for our literature review in orthopaedics and related fields.

Page: 1, Line: 40

Page: 2, Line: 44

Page: 3, Line: 56

Page: 3, Line: 59

Reviewer 2 Report

Dear authors,

This is a very interesting and valuable article, since the treatment to the patient is very important, and even more if he/she presents autism.

I propose some suggestions for improvement in some sections.

- The title is appropriate. It allows to quickly understand the purpose of the study.

- The abstract adequately presents a brief introduction, the aim of the research, the method, the results and main conclusions.

- The keywords are consistent.

- In the first paragraph, it is suggested to update the citation to APA 2022, as the DSM-V-RT (updated version) has been released.

- The introduction is really short. It is necessary to expand it further and justify the need for the study more extensively. Are there previous studies related to the topic or related to a similar field of research?

- Table 1 should be placed next to the text describing it, not in the Method section.

- Section 2 should be more complete and the PRISMA method should also be mentioned.

- Table 2 does not follow the MDPI standard. It should be modified.

- Point 3 is adequately developed and all relevant studies are cited.

- In the discussion, the most salient findings are adequately presented and the most relevant studies are cited.

- Sufficient current articles are cited and referenced.

- The References section should be improved, as the MDPI citation guidelines are not applied.

Author Response

We greatly appreciate the opportunity to revise our manuscript based on the recommendations of the reviewers. We have addressed each comment point by point below and have noted where changes appear in the clean, non-annotated manuscript by page and line number.

  1. The title is appropriate. It allows to quickly understand the purpose of the study.

We appreciate the reviewer’s assessment—no changes are necessary.

  1. The abstract adequately presents a brief introduction, the aim of the research, the method, the results and main conclusions.

We appreciate the reviewer’s assessment—no changes are necessary.

  1. The keywords are consistent.

We appreciate the reviewer’s assessment—no changes are necessary.

  1. In the first paragraph, it is suggested to update the citation to APA 2022, as the DSM-V-RT (updated version) has been released.

We appreciate the reviewer’s suggestion and have updated this reference in the introduction.

Page: 1, Line: 36

  1. The introduction is really short. It is necessary to expand it further and justify the need for the study more extensively. Are there previous studies related to the topic or related to a similar field of research?

We agree with the reviewer’s comment that our justification for more extensive study in the field of orthopaedics should be expanded. We have added additional text to the introduction to support this claim. We also have now included a reference to a similar review done in the emergency medicine setting that yielded valuable interventions for the emergency care setting. The objective of our review is to provide a similar review of the orthopaedic and related literature, with the hope of highlighting interventions specific to settings common in the movement sciences.

Page: 2, Line: 48

Page: 3, Line: 56

Page: 3, Line: 64

  1. Table 1 should be placed next to the text describing it, not in the Method section.

We have made this change in the manuscript.

Page: 2, Line: 44

  1. Section 2 should be more complete and the PRISMA method should also be mentioned.

We have further clarified the terms of our literature search and confirmed our use of the PRSIMA method when performing our review. We have also specified the team members involved in each of the steps of the screening and analysis process. We would be happy to include any other details that may make our Section 2 more complete.

Page: 3, Line: 80

Page: 3, Line: 84

Page: 3, Line: 89

  1. Table 2 does not follow the MDPI standard. It should be modified.

Table 2 has been modified to follow the MDPI standards, with vertical lines removed and only horizontal lines remaining.

Page: 5, Line: 120

  1. Point 3 is adequately developed and all relevant studies are cited.

We appreciate the reviewer’s assessment—no changes are necessary.

  1. In the discussion, the most salient findings are adequately presented and the most relevant studies are cited.

We appreciate the reviewer’s assessment—no changes are necessary.

  1. Sufficient current articles are cited and referenced.

We appreciate the reviewer’s assessment—no changes are necessary.

  1. The References section should be improved, as the MDPI citation guidelines are not applied.

The MDPI citation style guide has now been implemented in the References section.

Round 2

Reviewer 2 Report

Dear authors,

The article has improved after the changes made.

The proposed suggestions have been followed.